# Health worker acceptability of an HIV testing mobile health application within a rural Zambian HIV treatment programme

Andrés Montaner[1,2]*, Mulundu Mumbalanga[3], Marie-Chantal Umuhoza[3], Constance Wose Kinge[1,2], Emeka Okonji[1], Godfrey Ligenda[4], Eula Mothibi[1], Ben Chirwa[3], Pedro Pisa[1,5], Charles Chasela[1,2]

1 Right to Care, Centurion, South Africa, 2 Department of Epidemiology and Biostatistics, School of Public Health, Faculty of Health Sciences, University of the Witwatersrand, Johannesburg, South Africa, 3 Right to Care Zambia, Action HIV, Lusaka, Zambia, 4 United States Agency for International Development (USAID), Lusaka, Zambia, 5 Department of Human Nutrition, University of Pretoria, Pretoria, South Africa

* AndresM@gmail.com

## Abstract

### Background

As more people living with HIV are identified and prescribed antiretroviral treatment in Zambia, detecting new HIV infections to complete the last mile of epidemic control is challenging. To address this, innovative targeted testing strategies are essential. Therefore, Right to Care Zambia developed and implemented a novel digital health surveillance application, Lynx, in three Zambian provinces—Northern, Luapula, and Muchinga in 2018. Lynx offers real-time HIV testing data with geo-spatial analysis for targeted testing, and has proven effective in enhancing HIV testing yield. This cross-sectional mixed methods study assessed the acceptability of Lynx among HIV testing healthcare workers in Zambia.

### Methods

A quantitative Likert scale (1–5) survey was administered to 176 healthcare workers to gauge Lynx's acceptability. Additionally, six qualitative key person interviews and five focus group discussions were conducted to gain an in-depth understanding of acceptability, and identify relevant barriers and facilitators. Quantitative data were analysed by averaging survey responses and running descriptive statistics. Qualitative data were transcribed and analysed in thematic coding. Data triangulation was utilised between the data sources to verify findings.

### Results

Overall, the average survey score of perceived ease of use was 3.926 (agree), perceived usefulness was 4.179 (strongly agree) and perceived compatibility was 3.574

**Data availability statement:** All relevant data are within the paper and its Supporting Information files.

**Funding:** The author(s) received no specific funding for this work.

**Competing interests:** The authors have declared that no competing interests exist.

**Abbreviations:** AIDS, Acquired Immunodeficiency Syndrome; ART, Antiretroviral Treatment; GIS, Global Information System; HIV, Human Immunodeficiency Virus; KPI, Key Person Interview; NGO, Non-Government Organisation; PEPFAR, President's Emergency Plan for AIDS Relief; RTCZ, Right to Care Zambia; TAM, Technology Acceptance Model.

(agree). Survey questions related to network requirements, resource availability, and IT support had the most "strongly disagree" responses. The qualitative data collection revealed that Lynx was perceived as useful, and easy to use. Training for staff and regular updates were identified as facilitators, while conflicting work priorities and inconsistent IT support were identified barriers.

## Conclusion

Lynx was identified as acceptable by health workers due to its perceived usefulness, staff trainings, and regular updates. For a mobile health intervention to be embraced in rural Zambian settings, key facilitators include robust IT support, comprehensive training, user feedback-based updates, and consideration of facility staff priorities.

## Introduction

As of 2021, the global burden of HIV remains significant, with an estimated 37.7 million people living with the virus worldwide, and the African region continues to be the most affected, accounting for approximately two-thirds of new cases [1,2]. Despite substantial progress in managing the HIV epidemic, more than 8.1 million individuals remain unaware of their HIV status, contributing to ongoing transmission and mortality, with 1.5 million new infections reported in 2020 [3,4]. This underscores the critical importance of expanding HIV testing efforts.

Zambia has demonstrated remarkable strides in aligning with Joint United Nations Programme on HIV/AIDS (UNAIDS) goals, particularly the ambitious 95-95-95 targets, aiming for 95% of all people living with HIV to be aware of their status, 95% of diagnosed individuals to receive sustained antiretroviral therapy (ART), and 95% of those on ART to achieve viral suppression by 2025 [5,6]. In 2021, 88.7% of adults living with HIV in Zambia were aware of their status, with 98% on ART and 96.3% achieving viral suppression [7]. To reach the final milestone in identifying HIV-positive individuals, increasing testing has been identified as a pivotal strategy [8,9].

While mass testing and community outreach strategies have been instrumental in managing the HIV epidemic in Zambia, novel and efficient innovations such as index tracing and moonlighting are now being deployed by HIV treatment organisations to access harder-to-reach populations [10]. The President's Emergency Plan for AIDS Relief (PEPFAR) funded Zambian HIV treatment program, Right to Care-Zambia (RTCZ) has implemented the "Lynx" intervention, a mobile health (mHealth) application designed for targeted HIV testing. This application furnishes the HIV treatment programme with real-time HIV testing data aggregations, individual staff performance metrics, and geographic information systems (GIS) maps pinpointing targeted populations.

However, the success of such technological interventions hinges on the acceptance or "willingness to use" of their target users. Therefore, accounting for and ensuring high acceptability is crucial for maximising the application's potential in improving targeted HIV testing efficiency [11]. This study embedded within the RTCZ

program aims to evaluate the acceptability of the Lynx HIV testing mobile application among healthcare workers within a rural HIV treatment programme spanning three Zambian provinces. Acceptability is assessed through users' perceived ease of use, usefulness, and compatibility, while also investigating barriers and facilitators to inform future implementation considerations. The goal is to enhance our understanding of how healthcare providers engage with and perceive this innovative tool, optimising its integration into the healthcare system for more effective HIV testing strategies.

## Materials and methods

### Design

The study employed a mixed-method, cross-sectional design, integrating both quantitative and qualitative research components involving public health workers. In the quantitative phase, a survey was utilised, adapted from a well-established and validated tool known as the technology acceptance model (TAM) [12,13]. This tool comprises closed-ended questions designed to assess the acceptability of a given intervention, considering factors such as perceived ease of use, usefulness, and compatibility within a specific context. This survey draws on similar methodologies previously employed for assessing mHealth applications in sub-Saharan Africa, establishing its appropriateness for this study [14].

Complementing the quantitative approach, the qualitative component involved semi-structured key person interviews (KPI) and focus group discussions (FGD). These qualitative methods aimed to delve deeper into participants' perceptions and provide a richer and more nuanced understanding of their experiences with the intervention. The ensuing analysis incorporated data triangulation, reconciliation and validation of findings obtained from both quantitative and qualitative data collection methods.

This comprehensive mixed-method design ensures a multifaceted exploration of the acceptability of the intervention among public healthcare workers, offering a more holistic perspective that combines quantitative metrics with qualitative insights. Such an approach enhances the study's robustness and facilitates a more thorough comprehension of the complex dynamics surrounding the acceptance of the mHealth intervention in this public health context.

### Intervention assessed

In pursuit of optimising HIV testing programme resources through targeted testing, RTCZ has introduced the innovative HIV testing mHealth application Lynx in Zambia. This intervention is strategically aligned with the initial goal of the 95-95-95 targets, specifically focusing on HIV awareness. Lynx is meticulously tailored to adhere to the National Zambian HIV testing guidelines, serving as a comprehensive tool for both healthcare facilities and community-based HIV testing staff [15]. The Lynx application is installed on mobile tablets and, functioning offline, digitally captures detailed patient testing information, including demographics, sexually transmitted disease (STD) and tuberculosis (TB) screening, GIS coordinates, client care point modality, HIV test kit details, HIV test results, and the completion time for testing and counselling.

While connected to network the captured HIV testing data is instantaneously aggregated in a central online data warehouse, facilitating data review and analysis by the programme staff. If network is lost data is captured locally on the device and sent once connection is restored. Local staff engage in regular assessments, scrutinising demographic trends among HIV testing clients, evaluating facility entry points, and appraising facility staff performance. Additionally, geospatial coordinates are recorded at the point of HIV tests and complied to create timely hotspot maps for local health workers to review current testing coverage and emerging trends of new positive cases within their communities. The overarching aim of this mHealth intervention is to enhance operational efficiency by streamlining the processing of in-depth patient information, thereby maximising available resources within Zambia.

Intervention sites were selected based on the HIV testing performance of facilities within the broader HIV treatment programme. Among the 168 HIV treatment facilities covered by the programme, a subset of 55 "priority facilities" emerged, contributing a significant 85% of the total newly identified HIV-positive clients. The mHealth intervention has been

strategically implemented across all 55 priority facilities, distributed across Luapula (17 facilities), Muchinga (21 facilities), and Northern (17 facilities) provinces. To ensure seamless integration, all HIV testing and treatment staff in these priority facilities have undergone thorough training and have been equipped with the mHealth application, paving the way for comprehensive utilisation.

## Setting

RTCZ's HIV care and treatment initiatives operate actively in the northeast region of Zambia, spanning Luapula, Muchinga, and Northern provinces. This geographical area is characterised by its rural nature, sparse population, con-strained resources, and limited network coverage. It is situated approximately 14 hours by car from the nation's capital, Lusaka. Within this region, the HIV programme, serving as the backdrop for the research, extends support to a total of 168 public health HIV treatment facilities. These facilities encompass both primary and secondary care levels, collectively contributing to the comprehensive healthcare services delivered in this challenging and resource-limited environment.

## Participant selection

Data were collected at purposively sampled HIV treatment facilities. The participant population are comfortable in reading and speaking English, as English is the professional language of RTCZ. Study participation invitations were first sent to facility heads and managers of the relevant health workers for approval. All HIV testing healthcare workers using the HIV testing application were invited in English to participate in the survey over email from January 1st to February 28th, 2023. Of the 226 participants eligible, 176 completed the survey (77.88% response rate).

One HIV testing staff manager from two new random lottery selected facilities was invited in English from each province for six planned KPIs in November 2022, though only five were completed due to the unavailability of staff. The largest facility of each province was invited to partake in the FGDs in November 2022. Mansa General Hospital was invited in Luapula province, Nakonde Urban Health Center was invited in Muchinga province, and Mpulungu Urban Health Center was invited in Northern province. Each discussion included all mHealth intervention staff from the facility. Three additional FGDs were conducted during data collection to ensure data saturation of the qualitative themes. Saturation was deter-mined based on the recurring themes identified in the data collections. The total number of participants for each collection method is outlined in Table 1.

## Data collection

**Quantitative.**  All identified survey participants received a link to the online survey. The survey was anonymised, and self-administered in English on a user-friendly platform, REDCap. All healthcare workers had access to a computer with internet. The survey consisted of 15 statements written in plain English, four on the ease of use, four on usefulness, and seven on compatibility (S1 File). Participants were asked to rate their level of agreement on a 5-point Likert scale, where one was "strongly disagree" and five was "strongly agree". Before data collection, the survey was reviewed and adapted by five local Zambian expert staff with experience in HIV care and treatment to ensure that the survey questions were comprehendible and perceived clearly. The adapted survey was then pilot tested with five users of the application from a randomly selected facility. Following the piloting, the participants regarded the tool as comprehensible, and relevant, and perceived the questions.

**Table 1. Sample sizes of data collection.**

| Data Collection Tool | Sample Size |
|---|---|
| Surveys (Quantitative) | 176 persons |
| Key Person Interviews (Qualitative) | 5 persons |
| Focus Group Discussions (Qualitative) | 6 groups (31 persons) |

**Qualitative.** A semi-structured interview and group discussion guide (S2 and S3 Files) was developed to guide the KPIs and FGDs. Questions focused on participants' perceptions relating to barriers and facilitators of the application's compatibility, and ease of use and usefulness. KPIs and FGDs were conducted in English, on-site at the participant's facility and took 20–40 minutes. Before data collection, the interview guide was reviewed and adapted by five local Zambian experts with experience in HIV care and treatment to ensure the interview questions were comprehendible and perceived clearly. The reviewed interview guide was then pilot tested with two users of the application from a randomly selected facility.

## Data analysis

Survey data were inputted online through the REDCap platform and subsequently extracted into an Excel file for cleaning and analysis using R. Internal reliability and validity of the survey questions were assessed using Cronbach's alpha. The analysis focused exclusively on participants who had utilised the mHealth intervention (144 out of 176), with outliers identified and removed through z-scores. Utilising a Likert scale ranging from one to five, response averages fell within the range of 1.00 to 4.99. Each question received a mean score with standard deviations recorded, and responses were categorised into "agree" (averages 3.00–3.99), "strongly agree" (averages 4.00–4.99), "disagree" (averages 2.00–2.99), and "strongly disagree" (averages 1.00–1.99).

The questions within each factor group were further averaged, accounting for standard deviations. To assess the significance of participant demographics on acceptability factors, Kruskal-Wallis and Wilcoxon rank sum tests were employed.

As for qualitative data, the KPIs and FGDs transcriptions underwent coding in MAXQDA, utilising both inductive and deductive coding methods based on themes related to perceived usefulness, ease of use, and compatibility. Codes were applied to segments of the text representing the beginning and conclusion of relevant topics or transition to another subject. These codes were then organised into themes associated with the acceptability factors. Data triangulation, comparing findings from each data collection tool, was employed to validate key insights and ensure consistency across the diverse data sources. This robust approach enhances the reliability and credibility of the study's findings.

## Ethical considerations

This study was approved by the University of the Witwatersrand Human Research Ethics Committee (Medical) (Clearance certificate number M220720) and ERES Converge IRB in Lusaka Zambia (Ref No. 2021-Oct-003). All participants were over 18 years of age, implied consent was utilized for the anonymous online survey, and written consent was collected for the KPIs and FGDs. All transcripts have been deidentified for analysis and publication.

## Inclusivity in global research

Additional information regarding the ethical, cultural, and scientific considerations specific to inclusivity in global research is included in S8 File.

## Results

### Quantitative

**Survey participant characteristics.** Cronbach's Alpha of the 15-item survey was 0.663. Table 2 outlines the socio-demographic characteristics of the survey participants, the majority of whom are women, who came from Luapula and Northern Provinces, and were aged 25–34.

**Acceptability of Lynx.** Each statement has an averaged response score between 1.00 and 4.99. As shown (Table 3), most of the responses averaged 3.00–3.99 "agree" followed by 4.00–4.99 "strongly agree". A review of the frequency of the strongest scores, "strongly agree/disagree", in the responses indicated that over 10% of respondents strongly agreed

**Table 2. Demographic characteristics of survey participants.**

| | Luapula N (%) | Muchinga N (%) | Northern N (%) | Total N (%) |
|---|---|---|---|---|
| **Sex** | | | | |
| Men | 23 (33%) | 3 (21%) | 24 (39%) | 50 (35%) |
| Women | 46 (66%) | 11 (79%) | 37 (61%) | 94 (65%) |
| **Age Group** | | | | |
| 18–24 | 4 (1%) | 0 (0%) | 2 (3%) | 6 (4%) |
| 25–34 | 23 (33%) | 7 (50%) | 24 (39%) | 54 (38%) |
| 35–44 | 13 (19%) | 2 (14%) | 13 (21%) | 28 (19%) |
| 45–54 | 16 (23%) | 2 (14%) | 10 (16%) | 28 (19%) |
| 55+ | 13 (19%) | 3 (21%) | 12 (20%) | 28 (19%) |
| **Total** | | | | |
| | 69 (48%) | 14 (10%) | 61 (42%) | 144 |

**Table 3. Survey responses.**

| | Acceptability Factor | Question | Averaged (1–5) (SD) | Frequency of 1 "Strongly Disagree" (%n) | Frequency of 5 "Strongly Agree" (%n) |
|---|---|---|---|---|---|
| 1 | Perceived Ease of Use | Lynx is easy to use | 4.242 (0.430) | 0 (0%) | 31 (21.5%) |
| 2 | Perceived Ease of Use | It was (not*) hard for me to learn to use Lynx | 3.590* (1.132) | 8*(5.5%) | 21*(14.6%) |
| 3 | Perceived Ease of Use | The application layout is simple when moving between questions | 4.16 (0.368) | 0 (0%) | 20 (13.9%) |
| 4 | Perceived Ease of Use | Whenever I made a mistake using Lynx, I could correct the mistake quickly and easily | 3.721 (0.956) | 1 (0.7%) | 20 (13.9%) |
| 5 | Perceived Usefulness | Lynx is useful for my HIV testing practice | 4.233 (0.425) | 0 (0%) | 28 (19.4%) |
| 6 | Perceived Usefulness | Lynx helped me manage my patient's health effectively | 4.000 (0.790) | 0 (0%) | 28 (19.4%) |
| 7 | Perceived Usefulness | Lynx improved my access to delivering health care [sic] services | 4.175 (0.479) | 0 (0%) | 26 (18.1%) |
| 8 | Perceived Usefulness | Lynx has provided a helpful way to deliver healthcare services | 4.129 (0.428) | 0 (0%) | 19 (13.2%) |
| 9 | Perceived Compatibility | I could use Lynx even when the internet connection was poor or not available | 2.188 (1.085) | 33 (22.9%) | 4 (2.8%) |
| 10 | Perceived Compatibility | I think that Lynx fits well with the way I work | 3.940 (0.794) | 0 (0%) | 21 (14.6%) |
| 11 | Perceived Compatibility | I have the resources necessary to use Lynx | 3.407 (1.207) | 12 (8.3%) | 14 (9.7%) |
| 12 | Perceived Compatibility | I have the knowledge necessary to use Lynx | 4.202 (0.447) | 0 (0%) | 24 (1.67%) |
| 13 | Perceived Compatibility | Lynx is not compatible (is compatible*) with the way I work | 3.712* (0.962) | 1* (0.7%) | 18* (12.5%) |
| 14 | Perceived Compatibility | A specific person (or group) is available for assistance if I have difficulties concerning Lynx | 3.661 (1.134) | 11 (7.6%) | 18 (12.5%) |
| 15 | Perceived Compatibility | If I had the opportunity, I prefer (not*) working on paper | 3.492* (1.076) | 4*(2.8%) | 13* (9%) |

\* Response scale of the question (2, 13, 15) was inverted to align with the overall scale of "5/strongly agree" indicating higher acceptability.

with 11/15 statements, and over 10% of respondents strongly disagreed with 1/15 statements. These scores indicate that the intervention was perceived as easy to use and useful to the HIV testing staff, but there were factors, which could improve the compatibility with the context. These included issues related to statements 9, 11, and 14 of Table 3. Statement 9, relating to reliance on network connectivity, had the lowest response average of all questions, and the most "strongly disagree" responses. Statements 11 and 14 relating to healthcare workers' access to adequate technological resources and support to complete the application at work also scored lower.

Overall, statements under the "perceived usefulness" category had the most "strongly agree" responses, as well as slightly higher overall response averages as seen in Table 4.

Table 5 visualises the intervention's three acceptability factors by province. All factors scored over three, "agree" and "strongly agree". Yet, across all provinces, the perceived compatibility scored the lowest. Muchinga had the highest scores but also had the smallest sample size (n = 14). The data set failed the Shapiro-Wilk test for normality resulting in the use of medians, and the Kruskal-Wallis revealed none of the provinces to be significant on the acceptability factors.

Both genders matched the trend of lower perceived compatibility of the intervention compared to ease of use and usefulness as seen in Table 6. The Wilcoxon revealed neither gender to be significant on the outcomes.

Table 7 indicates a tendency among all age groups to score perceived compatibility lower, but 18–24 also had lower scores overall. It is worth noting that 18–24 also had the smallest group sample size (n = 6). The Kruskal-Wallis revealed none of the age groups to be significant on the outcomes.

**Table 4. Averaged acceptability factor scores.**

| Characteristic | N = 144[1] |
|---|---|
| Perceived Ease of Use | 3.926 (*0.436*) |
| Perceived Usefulness | 4.179 (*0.422*) |
| Perceived Compatibility | 3.574 (*0.467*) |

[1] Mean (*SD*).

**Table 5. Survey findings by province.**

| Characteristics | Province | | | p-value[2] |
|---|---|---|---|---|
| | Luapula, *N = 69*[1] | Muchinga, *N = 14*[1] | Northern, *N = 61*[1] | |
| Perceived Ease of Use | 4.000 (*0.500*) | 4.000 (*0.500*) | 4.000 (*0.500*) | 0.089 |
| Perceived Usefulness | 4.000 (*0.500*) | 4.375 (*0.500*) | 4.000 (*0.000*) | 0.11 |
| Perceived Compatibility | 3.714 (*0.429*) | 3.857 (*0.429*) | 3.571 (*0.536*) | 0.5 |

[1] Median (*IQR*).

[2] Kruskal-Wallis rank sum test.

**Table 6. Survey findings by gender.**

| Characteristics | Gender | | p-value[2] |
|---|---|---|---|
| | Male, *N = 50*[1] | Female, *N = 94*[1] | |
| Perceived Ease of Use | 4.000 (*0.500*) | 4.000 (*0.500*) | 0.7 |
| Perceived Usefulness | 4.000 (*0.750*) | 4.000 (*0.250*) | 0.5 |
| Perceived Compatibility | 3.714 (*0.571*) | 3.571 (*0.571*) | 0.5 |

[1] Median (*IQR*).

[2] Wilcoxon rank sum test.

**Table 7. Survey findings by age.**

| Characteristics | Age Group | | | | | *p*-value[2] |
|---|---|---|---|---|---|---|
| | 18–24, N=6[1] | 25–34, N=54[1] | 35–44, N=28[1] | 45–54, N=28[1] | 55+, N=28[1] | |
| Perceived Ease of Use | 3.875 (*0.313*) | 4.000 (*0.500*) | 4.000 (*0.500*) | 4.000 (*0.500*) | 4.000 (*0.500*) | 0.5 |
| Perceived Usefulness | 3.875 (*0.438*) | 4.000 (*0.250*) | 4.000 (*0.000*) | 4.000 (*1.000*) | 4.000 (*0.500*) | 0.14 |
| Perceived Compatibility | 3.357 (*0.286*) | 3.714 (*0.429*) | 3.571 (*0.571*) | 3.500 (*0.714*) | 3.571 (*0.643*) | 0.4 |

[1] Median (*IQR*).

[2] Kruskal-Wallis rank sum test.

### Qualitative

Six themes were identified concerning the acceptability of the Lynx intervention. These included efficiency gains, training, network/IT, feedback-based updates, community vs facility, and work priorities.

**Efficiency gains.** The KPI and FGD findings revealed an overall perception that the Lynx innovation was easy to use and incredibly useful for a range of work benefits. The main healthcare uses highlighted were the live reporting function and the production of local GIS HIV testing maps. The HIV testing staff noted that Lynx allows for live programme reporting through the tablet, which saves data recording and submission times, and enables managers to see each staff member's work performance in real-time.

*"it [sic] has simplified matters because if we are using hard copies to send this information it would take a long time to reach you, and maybe in the process we lose [sic] the documents but this is instant, so it's good" – Kasama General health worker.*

*"Its [sic] not (difficult) …… because paperwork is tedious and using paperwork you can easily lose the information" – Mpika Urban health worker.*

*"And the paper will pass through many people, our supervisor, the district in the process it may get lose [sic] and things like that so redo this, but this is just direct I think it's good" – Kasama General health worker.*

*"well [sic] no it's a good tool, it was effective, at time [sic] it could give us accurate information that we could get better results" – Luwingu District Hospital health worker.*

**Training.** While most participants perceived Lynx as easy to complete, it was clear that comprehensive training facilitated the HIV testing staff's ease of use of the application. Those who received training appreciated the capacity building and those who did not receive training expressed their desire to be trained.

*"For the first time it was quite difficult for me but when I got used to it, it was very nice" – Tazara Rural Health Clinic health worker.*

*"Lynx it is easy, so what can I say" – Lubwe Mission Hospital health worker.*

*"No no we did not train, I think that was the biggest challenge more [sic] especially with me because I am not too conversant with smart phones so some applications where [sic] difficult to use" – Mpika Urban health worker.*

**Network and IT.** Some participants faced a barrier to using Lynx due to network and IT challenges that come with rural community work. It was noted that IT repair times varied and could prevent HIV testing staff from using Lynx for weeks at a time.

*"For me since I was always based at the facility it was easy because of network [sic], except for the counsellors that [sic] were going in the field. Yes we have a bit of a challenge because you find that they have to be there and it can be*

*something where you test somebody and you record but then you find that sometimes you don't have network [sic]., so, it's a bit of a challenge" – Mpepo Rural Health Clinic health worker.*

*"I think for me as well especially when the tablet is ok, there is no problem" – Kasama Urban health worker.*

*"When they take it to the office [for repair], but for those people to bring it back, it takes ages. That's the only thing" – Tazara Rural Health Clinic health worker.*

*"Lynx has stopped working as right [sic] now so that I can make a request to the office so they can bring us in [sic] the normal system so that Lynx would start working again, that is the biggest challenge I have seen." – Lubwe Mission Hospital health worker.*

*"sometimes [sic] we need upgrading when you take these tablets to the office they are not done in time" – Kasama General health worker.*

*"he [sic] is just alone using the tablet so mostly he only captures the ones he has tested himself" – Chibansa Urban health worker.*

**Feedback-based updates.** Several HIV testing staff noted that a previous update of the application further facilitated Lynx's compatibility and ease of use in their work setting. The update was based on user feedback.

*"At first, it was a bit difficult to use Lynx, because it was too long and the questions were too many" – Chibansa Urban health worker.*

*"Now that the questions have been simplified and some parts have been updated, … it made the usage a little bit familiar and user friendly" – Chibansa Urban health worker.*

*"Today some of the questions have been removed. They are not necessarily because some are a duplication of information. But as it is now, it's ok" – Kasama Urban health worker.*

**Facility vs community.** In general, the tablet application was described as compatible with rural settings, but there were different opinions on whether Lynx was a better fit for the community or the facility. Some counsellors prefer using Lynx in a more structured facility as opposed to using it in the community.

*"so {sic} you don't have a place where you can properly work on the Lynx, unlike the facility, because at the facility obviously I will have a room where each client has privacy… and the client is siting [sic] comfortable [sic] as I enter. For the community its [sic] different" – Mpepo Rural Health Clinic health worker.*

By contrast, for some, it was the lack of disturbance in the community that made the use of the app more feasible in the community.

*"in [sic] the community we are free, there are no disturbances but here some we are disturbed by people coming in this room coming in, come knocking knocking" – Lubwe Mission Hospital health worker.*

Additional concerns revolved around patient's negative perceptions, and feelings of discomfort and distress with their information being captured into a tablet.

*"…where we are as a set up the clients would find it a bit rude when you are on the phone they would not understand that it's the application that I am using, yes so it was a bit difficult" – Mpepo RHC health worker.*

*"That is what I am saying I am saying it's the area, maybe in other places it might be different where these people are acquainted to [sic] electronics, but for [sic] here it's kind of difficult it's like you are on the phone you are not paying attention to the client (inaudible) hinder us in a way"* – Mpepo RHC health worker.

**Work priorities.** A few of the HIV testing staff noted that the application required too much time to complete in a busy facility setting, especially since the staff are required to double capture the data into paper registers and the Lynx tablet. This challenge was often brought up about the previously mentioned training and IT challenges.

*"what she is saying [sic] maybe we are not in the ward, there is [sic] a lot of patients whom I need to do, to test, now that is a challenge for me to enter those clients in the Lynx because it take [sic] 45 minutes for me to complete a session because what we were told is that we should take 45 minutes for one clients [sic], so for me I don't know"* – Mansa General Hospital health worker.

*"double [sic] capture which is more of time consuming [sic] in that way maybe Lynx could be consistently [sic] if we just use it that data are able to extract it from there of the same client"* – Luwingu District Hospital health worker.

*"when [sic] it's a busy day he doesn't enter directly on the Lynx application so that it quickens the process he would just use the hard copy to screen into and capture all the details that he needs to use"* – Chibansa Urban health worker.

*"especially with the facility like this there is too much work to do mostly we use registers"* – Kasama Urban health worker.

*"the [sic] only challenge that we have is time management, you know when you need to counsel a person and we need that information the client feels like you are delaying them, for us to do the exact way but the client feels you are delaying them. For an [sic] example if you ask this question you ask this one you ask this one, and they feel you are wasting their time"* – Mansa General Hospital health worker.

## Discussion

This study aimed to evaluate the acceptability of a targeted HIV testing mobile application within a Zambian rural HIV treatment programme. The results indicate that a substantial majority of the participating HIV testing staff found the mHealth intervention acceptable within their setting. Survey findings underscored the strong perception of the intervention's usefulness, ease of use, and compatibility with the demands of a rural HIV testing environment. Barriers and facilitators influencing the application's use and overall acceptability were identified. Notably, no significant differences were observed between provinces, age groups, or genders among survey participants.

Qualitative data echoed the perceived usefulness of the intervention, emphasising the theme of efficiency gained through its implementation. While the GIS capability was acknowledged, the clear benefit highlighted by the intervention users was the live digital data-capturing feature. Existing studies on data digitalisation and real-time reporting in healthcare underscore the efficiency gains associated with such technological advancements [16–18]. Participants valued the instantaneous review of individual performance and the efficiency of digital data submission for HIV testing programme data.

The ease of use of the mHealth intervention was acknowledged by survey participants, aligning with the broader trend of healthcare workers in sub-Saharan Africa finding mobile-based interventions accessible [19]. That said, trainings emerged as a critical facilitator to ensure all the staff, regardless of experience, were adept with the mHealth intervention. Comprehensive staff training is a well-established factor in the success of mHealth interventions, with repeated training sessions over time being suggested in some studies [20]. Additionally, the study participants valued the user-based updates to the software, ensuring the intervention's adaptation to their evolving contextual and work requirements.

Perceived compatibility differed between facility and community settings, some participants expressing a preference for the structured environment of a facility, while others favoured the more flexible community work setting. In both settings instances were reported where Lynx use could make HIV testing clients uncomfortable or feel judged due to the introduction of new technology unknown to the client, particularly in rural communities. These compatibility finding suggests a need for nuanced strategies to adapt the intervention to specific settings, or providing additional training to address specific contexts within the facilities and communities. To account for client perception of mHealth interventions, additional research is needed into the perception of mHealth interventions by rural sub-Saharan health clients.

Barriers identified related to IT challenges and competing priorities within facilities. IT support, particularly issues related to equipment maintenance, power availability, and internet availability, surfaced as significant barriers. This emphasises the importance of comprehensive IT support systems to overcome technical challenges and ensure sustained use. The study highlights the need for strategic placement of trained personnel onsite to assist with common IT challenges, particularly in settings where technical knowledge for maintenance remains limited despite the widespread adoption of mobile technology in Africa [19,21,22]. Some staff did not consistently prioritise the time to complete the application due to multiple demands during a work shift and multiple data capturing requirements on different platforms. Given the dynamic work environment of health facilities, practical clear delegation of responsibilities is needed to ensure all duties are fulfilled as planned. Additionally, the delay due to multiple data capturing demonstrates a larger need for program data streamlining and interoperability, which is a recognized challenge in the region [23,24]. As one of Lynx's precepted benefits is the instantaneous data submission and review, its data streamlining capabilities can be further reviewed to address this challenge.

Fortunately, it has been found that population familiarity of mobile technology, and mobile phones, is a driver for sustainable mHealth intervention scalability in Africa. When looking to sustainably scale mHealth interventions in Zambia ease of use, perceived benefits, integration into program context, and comprehensive management are key themes identified for successful scalability [19]. Therefore, the scalability of Lynx is dependent on continued integration into the program with user feedback-based updates for ease of use and perceived benefits, comprehensive context specific trainings of all intervention users, concrete IT support from roving or locally trained staff, and clear work priorities and designated responsibilities for intervention staff.

The study provides valuable insights into the acceptability of the targeted HIV testing mobile application within a Zambian rural HIV treatment programme. The findings emphasise the multifaceted nature of the challenges and opportunities associated with mHealth interventions, suggesting the importance of tailored strategies and ongoing support to enhance their effectiveness and acceptance within diverse healthcare settings.

## Strengths and limitations

This study stands out due to its comprehensive approach, encompassing both quantitative user acceptability surveys and qualitative individual and group acceptability data to gauge the acceptability of an mHealth intervention. By employing a survey, mHealth users directly assessed acceptability, while KPIs and FGDs provided valuable contextual insights through the sharing of lived experiences.

However, the reliance on self-reported findings introduces potential biases, including selective memory, attribution errors, or exaggeration. Social desirability bias, influenced by participants' involvement in the HIV programme, may contribute to more positive feedback. Furthermore, the study is constrained by its language of data collection, as multiple dialects, including English, are spoken by the target population. Although English is widely spoken in the area, the decision to conduct data collection solely in English may introduce bias related to language fluency. Pilot testing of the data collection tools and participants occasionally translating for one another during discussions assisted in mitigating any language barrier.

An additional limitation lies in the study's exclusive focus on health workers, omitting the perspectives of mHealth intervention clients. The absence of client perceptions poses a potential gap in the overall understanding of the intervention's acceptability within the broader context of its intended beneficiaries.

## Conclusion

The Lynx HIV testing application received favourable perceptions from HIV testing staff, who found it easy to use, compatible with a rural HIV programme, and incredibly useful. This study underscores the significance of providing comprehensive training for all users and emphasises the crucial role of feedback-based updates to address evolving programme needs and user contexts. While the intervention has gained acceptance among its targeted users through effective training and updates, challenges persist, particularly concerning competing work priorities and consistent IT support.

Although acceptability is conventionally regarded as a user's perception and willingness to use an intervention, this study illuminates how acceptability levels can be influenced by contextual barriers or facilitators. Understanding these dynamics is vital for RTCZ and other sub-Saharan Africa HIV treatment programmes to navigate challenges and leverage facilitators effectively, enhancing the acceptability of mHealth HIV testing interventions among program staff. By incorporating these findings on the barriers and facilitators of acceptability, the RTCZ programme can tailor strategies to boost acceptability levels, leading to improved outcomes in the implementation of the mHealth HIV testing intervention.

### Definition of terms

| | |
|---|---|
| **mHealth**: Mobile health | |
| **Modality**: Health facility care entrance point leading to HIV test (e.g., paediatric ward, antenatal care ward, inpatient ward, outpatient ward, index client intake) | |
| **Tablet**: Mobile device through which a mobile application is utilised | |
| **Yield**: % of positive tests in a given period (tests positive/total tests) | |

## Supporting information

**S1 File. Lynx acceptability survey.**
(DOCX)

**S2 File. Key person interview guide.**
(DOCX)

**S3 File. Focus group discussion guide.**
(DOCX)

**S4 File. University of Witwatersrand ethical clearance.**
(PDF)

**S5 File. ERES Converge Zambia original ethical clearance.**
(PDF)

**S6 File. ERES Converge Zambia renewal letter.**
(PDF)

**S7 File. ERES Converge Zambia renewed ethical clearance.**
(PDF)

**S8 File. Inclusivity in global research checklist.**
(DOCX)

**S9 File. Quantitative data.**
(ZIP)

**S10 File. Qualitative data.**
(ZIP)

## Acknowledgments

We express our gratitude to the study participants for their valuable contribution to the study as well as the Zambia Action HIV and Right to Care Organisations for their support in the study design.

## Author contributions

**Conceptualization:** Andrés Montaner, Pedro Pisa, Charles Chasela.

**Formal analysis:** Andrés Montaner.

**Funding acquisition:** Godfrey Ligenda, Eula Mothibi, Ben Chirwa.

**Investigation:** Andrés Montaner, Mulundu Mumbalanga.

**Methodology:** Andrés Montaner, Constance Wose Kinge, Emeka Okonji, Pedro Pisa, Charles Chasela.

**Project administration:** Mulundu Mumbalanga, Marie-Chantal Umuhoza, Godfrey Ligenda, Ben Chirwa.

**Resources:** Marie-Chantal Umuhoza, Godfrey Ligenda, Eula Mothibi, Ben Chirwa.

**Supervision:** Constance Wose Kinge, Emeka Okonji, Pedro Pisa, Charles Chasela.

**Writing – original draft:** Andrés Montaner.

**Writing – review & editing:** Eula Mothibi, Pedro Pisa, Charles Chasela.

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
