## [Decision Letter · Decision Letter 0]

10 Jan 2025

PONE-D-24-43908Health worker acceptability of an HIV testing mobile health application within a rural Zambian HIV treatment programmePLOS ONE

Dear Dr. Montaner,

Thank you for submitting your manuscript to PLOS ONE. After careful consideration, we feel that it has merit but does not fully meet PLOS ONE’s publication criteria as it currently stands. Therefore, we invite you to submit a revised version of the manuscript that addresses the points raised during the review process.

Please note that we have only been able to secure a single reviewer to assess your manuscript. We are issuing a decision on your manuscript at this point to prevent further delays in the evaluation of your manuscript. Please be aware that the editor who handles your revised manuscript might find it necessary to invite additional reviewers to assess this work once the revised manuscript is submitted. However, we will aim to proceed on the basis of this single review if possible. 

The reviewers comments are available below, please review their assessment and make the appropriate revisions to your manuscript. 

We look forward to receiving your revised manuscript.

Kind regards,

Emma Campbell, Ph.D

Staff Editor

PLOS ONE

Journal Requirements:

3. We note you have included a table to which you do not refer in the text of your manuscript. Please ensure that you refer to Tables 1 and 2 in your text; if accepted, production will need this reference to link the reader to the Table.

4. Please include a copy of Tables 8 and 9 which you refer to in your text on page 12 on your PDF file.

5. We notice that your supplementary [S1 Lynx Acceptability Survey] are included in the manuscript file. Please remove them and upload them with the file type 'Supporting Information'. Please ensure that each Supporting Information file has a legend listed in the manuscript after the references list.

Reviewers' comments:

Reviewer's Responses to Questions

**Comments to the Author**

1. Is the manuscript technically sound, and do the data support the conclusions?

Reviewer #1: Yes

2. Has the statistical analysis been performed appropriately and rigorously? 

Reviewer #1: Yes

3. Have the authors made all data underlying the findings in their manuscript fully available?

Reviewer #1: Yes

4. Is the manuscript presented in an intelligible fashion and written in standard English?

Reviewer #1: Yes

5. Review Comments to the Author

Reviewer #1: Congratulations to the authors for this important piece of work. This innovation/research is very much needed to intensify HIV case finding, close the treatment gap for PLHIV and achieve epidemic control of HIV/AIDS in Zambia and other countries in sub-Saharan Africa. The mHealth application can potentially increase HIV testing coverage in Zambia and improve the efficiency of HIV testing services through hotspot mapping, real-time data collection and reporting, and HCW's performance measurement. Findings will support the design, adaptation, and scale-up of mHealth interventions in Zambia.

Comments

Abstract

- Nil

Introduction

- Line 95: UNAIDS write in full, first mention

Materials and Methods

• Since the list of study participants was first sent to facility heads for confirmation and approval to participate in surveys, IDI and FGDs, how did you address the possibility of selection and confirmation bias in your study? Facility heads may have selected ‘’best staff’’ to respond to survey and participate in interviews or FGDs

• What is the survey response rate among participants? What proportion of HIV testing healthcare workers using the mHealth application responded to the survey?

• What informed the minimum number of participants for IDI? How did you ensure data saturation?

• Considering that only participants from high-volume facilities were invited to partake in the FGDs and HIV testing managers for IDI, how did you ensure maximum variation to account for perspectives from the small facilities, probably in rural communities where there is little or no internet

o How will this intervention work in rural communities with junior healthcare workers who have less experience/education and limited internet access?

Findings

• Line 265: elaborate on the type of or unpack the resources statement 11 is referring to:’’ I have the resources necessary to use Lynx’’

• In-text table citations are incorrect for tables two to seven in the manuscript. Review all in-text citations and labelling for tables in the manuscript and ensure correct reference.

Discussion

• The authors acknowledged the capability of GIS while emphasising the usefulness of real-time data-capturing and reporting. Can authors elaborate on the production of local GIS HIV testing maps (GIS hotspots) as one of the healthcare uses, probably in their description of the intervention? Authors should elaborate on the type of geocordinates (residential or point of care/testing) collected that made GIS hotspots possible (pls note that this is optional). If only POC testing geocordinates are collected, discuss this implication for your programming from HCWs perspectives.

• Lines 423-427 and Lines 423 – 427 discuss the same theme. Understanding the different actors (for whom) and in what contextual conditions did the intervention (mHealth) work? Authors should expatiate to clarify or summarise the two paragraphs into one.

• Considering the concern about patient's negative perceptions and feelings of discomfort, future research that assesses the perception of clients/patients is needed

• Line 428 – 436: In a setting constrained by limited resources, providing mobile phones, maintaining and updating mobile applications, and building staff capacity can be challenging. Can you discuss the future of mobile health applications in Zambia in terms of scale-up and sustainability?

• What are the strengths and limitations of this study? discuss

6. PLOS authors have the option to publish the peer review history of their article (what does this mean? ). If published, this will include your full peer review and any attached files.

**Do you want your identity to be public for this peer review?** For information about this choice, including consent withdrawal, please see our Privacy Policy .

Reviewer #1: No

---

## [Author Response · Author response to Decision Letter 1]

7 Mar 2025

PONE-D-24-43908

Good day PLOS ONE Editor,

We were pleased receive your comments for Health worker acceptability of an HIV testing mobile health application within a rural Zambian HIV treatment programme. After going through the revisions, we are pleased present the improved manuscript for your consideration. In the section below each comment is addressed with direct revisions reflected in the newly submitted manuscript and supporting files.

First Editor

Line 95: UNAIDS write in full, first mention

Updated to “Joint United Nations Programme on HIV/AIDS”

Since the list of study participants was first sent to facility heads for confirmation and approval to participate in surveys, IDI and FGDs, how did you address the possibility of selection and confirmation bias in your study? Facility heads may have selected ‘’best staff’’ to respond to survey and participate in interviews or FGDs

Wording clarified in lines 173-1843“All HIV testing healthcare workers using the HIV testing application were invited in English to participate in the survey over email from January 1st to February 28th 2023.” There was no selection of workers for the survey, all staff who had used the intervention were invited to participate via email.

“One HIV testing staff manager from two new random lottery selected facilities was invited in English from each province for six planned KPIs in November 2022, though only five were completed due to the unavailability of staff.” The participant for the interviews was selected through random lottery, and there was no influence of the facility heads for participant selection.

“The largest facility of each province was invited to partake in the FGDs in November 2022. Mansa General Hospital was invited in Luapula province, Nakonde Urban Health Center was invited in Muchinga province, and Mpulungu Urban Health Center was invited in Northern province… Each discussion included all mHealth intervention staff from the facility.” All available staff at selected facilities participated in the focus group discussion, there was no selection of staff.

What is the survey response rate among participants? What proportion of HIV testing healthcare workers using the mHealth application responded to the survey?

Added response rate in line 175 “Of the 226 participants eligible, 176 completed the survey (77.88% response rate).”

What informed the minimum number of participants for IDI? How did you ensure data saturation?

6 IDI were planned in accordance with program budget approvals, travel between each facility required advance planning due to road conditions. Lines 177-179 “One HIV testing staff manager from two new random lottery selected facilities was invited in English from each province for six planned KPIs in November 2022, though only five were completed due to the unavailability of staff.”

Added line 184 “Saturation was determined based on the recurring themes identified in the data collections.”

Considering that only participants from high-volume facilities were invited to partake in the FGDs and HIV testing managers for IDI, how did you ensure maximum variation to account for perspectives from the small facilities, probably in rural communities where there is little or no internet

Line 170 “Data were collected at purposively sampled HIV treatment facilities.”

Lines 177 “One HIV testing staff manager from two new random lottery selected facilities was invited in English from each province for six planned KPIs.”

Due to budgeting approvals data were purposively sampled from the larger facilities with most use, however the IDIs/KPIs utilized random sampling.

Updated line 162 “This geographical area is characterised by its rural nature, sparse population, constrained resources, and limited network coverage.”

All of the facilities within the region face internet and resource challenges, the random sampling of the KPIs and the survey completion from all facilities allowed for input from the smaller facilities to be captured.

Intervention details updated to clarify use with limited internet access in lines 140-143 “While connected to network the captured HIV testing data is instantaneously aggregated in a central online data warehouse, facilitating data review and analysis by the programme staff. If network is lost data is captured locally on the device and sent once connection is restored.”

How will this intervention work in rural communities with junior healthcare workers who have less experience/education and limited internet access?

While experience/education was not captured in the data collection of this study, the theme of comprehensive staff training was identified as a facilitator. The theme speaks to addressing the challenges of mixed the experiences within a rural health worker cohort. Updated lines 407-411 to clarify “That said, trainings emerged as a critical facilitator to ensure all the staff, regardless of experience, were adept with the mHealth intervention. Comprehensive staff training is a well-established factor in the success of mHealth interventions, with repeated training sessions over time being suggested in some studies (20).”

Intervention details updated to clarify use with limited internet access in lines 140-143 “While connected to network the captured HIV testing data is instantaneously aggregated in a central online data warehouse, facilitating data review and analysis by the programme staff. If network is lost data is captured locally on the device and sent once connection is restored.”

Line 265: elaborate on the type of or unpack the resources statement 11 is referring to:’’ I have the resources necessary to use Lynx’’

Updated line 262 “Statements 11 and 14 relating to healthcare workers’ access to adequate technological resources, and support to complete the application at work…”

In-text table citations are incorrect for tables two to seven in the manuscript. Review all in-text citations and labelling for tables in the manuscript and ensure correct reference.

Table citations updated

The authors acknowledged the capability of GIS while emphasising the usefulness of real-time data-capturing and reporting. Can authors elaborate on the production of local GIS HIV testing maps (GIS hotspots) as one of the healthcare uses, probably in their description of the intervention? Authors should elaborate on the type of geocordinates (residential or point of care/testing) collected that made GIS hotspots possible (pls note that this is optional). If only POC testing geocordinates are collected, discuss this implication for your programming from HCWs perspectives.

Added lines 145-147 “Additionally, geospatial coordinates are recorded at the point of HIV tests and complied to create timely hotspot maps for local health workers to review current testing coverage and emerging trends of new positive cases within their communities.”

Lines 423-427 and Lines 423 – 427 discuss the same theme. Understanding the different actors (for whom) and in what contextual conditions did the intervention (mHealth) work? Authors should expatiate to clarify or summarise the two paragraphs into one.

Agreed, paragraphs revised and combined

Considering the concern about patient's negative perceptions and feelings of discomfort, future research that assesses the perception of clients/patients is needed

Agreed added line 420 “To account for client perception of mHealth interventions, additional research is needed into the perception of mHealth interventions by rural sub-Saharan health clients..”

Line 428 – 436: In a setting constrained by limited resources, providing mobile phones, maintaining and updating mobile applications, and building staff capacity can be challenging. Can you discuss the future of mobile health applications in Zambia in terms of scale-up and sustainability?

Added paragraph lines 440-448 to discuss scale-up and sustainability

What are the strengths and limitations of this study? Discuss

Strengths and limitations section added lines 452-468

Confirm PLOSONE style requirements

Headings and authorship format adjusted

Attach questionnaire on inclusivity in global research

Questionnaire included in submission as “S8 File. Inclusivity in global research checklist”

Include tables 8 and 9 on page 12 in PDF file

Tables 8 and 9, now tables 6 and 7 included as PDF files in submission files

We notice that your supplementary [S1 Lynx Acceptability Survey] are included in the manuscript file. Please remove them and upload them with the file type 'Supporting Information'. Please ensure that each Supporting Information file has a legend listed in the manuscript after the references list.

Supplementary information submitted under separated supplementary information files

Please review your reference list to ensure that it is complete and correct. If you have cited papers that have been retracted, please include the rationale for doing so in the manuscript text, or remove these references and replace them with relevant current references. Any changes to the reference list should be mentioned in the rebuttal letter that accompanies your revised manuscript. If you need to cite a retracted article, indicate the article’s retracted status in the References list and also include a citation and full reference for the retraction notice

Updated reference 6 “Frescura L, Godfrey-Faussett P, Feizzadeh A, El-Sadr W, Syarif O, Ghys PD, et al. Achieving the 95 95 95 targets for all: A pathway to ending AIDS. PLOS ONE. 2022 Aug 4;17(8):e0272405.” to the updated corrected publication as the authorship of the original article was amended.

Second Editor

We note that this data set consists of interview transcripts. Can you please confirm that all participants gave consent for interview transcript to be published? If they DID provide consent for these transcripts to be published, please also confirm that the transcripts do not contain any potentially identifying information (or let us know if the participants consented to having their personal details published and made publicly available).

I confirm that the participants consented to the publication of the transcripts and there is no potentially identifying information

Can you please upload an additional copy of your revised manuscript that does not contain any tracked changes or highlighting as your main article file.

Additional copy uploaded

Your ethics statement should only appear in the Methods section of your manuscript. If your ethics statement is written in any section besides the Methods, please delete it from any other section.

Ethics statement is only listed in the methods section under “ethical considerations,” lines 235-240, (removed statements related to consent from data collection sections)

---

## [Decision Letter · Decision Letter 1]

23 Apr 2025

Health worker acceptability of an HIV testing mobile health application within a rural Zambian HIV treatment programme

PONE-D-24-43908R1

Dear Dr. Montaner,

We’re pleased to inform you that your manuscript has been judged scientifically suitable for publication and will be formally accepted for publication once it meets all outstanding technical requirements.

Kind regards,

Guendalina Capece

Academic Editor

PLOS ONE

Additional Editor Comments (optional):

Reviewers' comments:

Reviewer's Responses to Questions

**Comments to the Author**

1. If the authors have adequately addressed your comments raised in a previous round of review and you feel that this manuscript is now acceptable for publication, you may indicate that here to bypass the “Comments to the Author” section, enter your conflict of interest statement in the “Confidential to Editor” section, and submit your "Accept" recommendation.

Reviewer #1: All comments have been addressed

2. Is the manuscript technically sound, and do the data support the conclusions?

Reviewer #1: Yes

3. Has the statistical analysis been performed appropriately and rigorously? 

Reviewer #1: Yes

4. Have the authors made all data underlying the findings in their manuscript fully available?

Reviewer #1: Yes

5. Is the manuscript presented in an intelligible fashion and written in standard English?

Reviewer #1: Yes

6. Review Comments to the Author

Reviewer #1: (No Response)

7. PLOS authors have the option to publish the peer review history of their article (what does this mean? ). If published, this will include your full peer review and any attached files.

**Do you want your identity to be public for this peer review?** For information about this choice, including consent withdrawal, please see our Privacy Policy .

Reviewer #1: No

---

## [Editor Report · Acceptance letter]

PONE-D-24-43908R1

PLOS ONE

Dear Dr. Montaner,

I'm pleased to inform you that your manuscript has been deemed suitable for publication in PLOS ONE. Congratulations! Your manuscript is now being handed over to our production team.

Kind regards,

on behalf of

Professor Guendalina Capece

Academic Editor

PLOS ONE